Automated lesion detection in cotton leaf visuals using deep learning

Akbar Frnaz 1
Aribi Yassine 2
Muhammad Usman Syed 3 drsyedmusman@gmail.com
Faraj Hamzah 2
Murayr Ahmed 2
Alasmari Fawaz 2
Khalid Shehzad 4
1 Department of Creative Technologies, Faculty of Computing and AI, Air University , Islamabad , Pakistan
2 Department of Science and Technology, College of Ranyah, Taif University , Taif , Saudi Arabia
3 Department of Computer Science, Bahria School of Engineering and Applied Sciences , Islamabad , Pakistan
4 Department of Computer Engineering, Bahria School of Engineering and Applied Sciences , Islamabad , Pakistan
Balas Valentina Emilia
Electronic publication date: 2024 Oct 18
Publication date: 2024
Volume: 10
Electronic Location ID: e2369
Received 2024 Jun 14; Accepted 2024 Sep 7
Copyright: © 2024 Akbar et al.
Copyright year: 2024
Copyright holder: Akbar et al.
License: This is an open access article distributed under the terms of the Creative Commons Attribution License, which permits unrestricted use, distribution, reproduction and adaptation in any medium and for any purpose provided that it is properly attributed. For attribution, the original author(s), title, publication source (PeerJ Computer Science) and either DOI or URL of the article must be cited.
License URL: https://creativecommons.org/licenses/by/4.0/

Keywords: Cotton disease detection, Deep learning, Feature fusion, Precision agriculture, CNN

Funding: The Deanship of Graduate Studies and Scientific Research, Taif University The Deanship of Graduate Studies and Scientific Research, Taif University supported this work. The funders had no role in study design, data collection and analysis, decision to publish, or preparation of the manuscript.

==============================
Cotton is one of the major cash crop in the agriculture led economies across the world. Cotton leaf diseases affects its yield globally. Determining cotton lesions on leaves is difficult when the area is big and the size of lesions is varied. Automated cotton lesion detection is quite useful; however, it is challenging due to fewer disease class, limited size datasets, class imbalance problems, and need of comprehensive evaluation metrics. We propose a novel deep learning based method that augments the data using generative adversarial networks (GANs) to reduce the class imbalance issue and an ensemble-based method that combines the feature vector obtained from the three deep learning architectures including VGG16, Inception V3, and ResNet50. The proposed method offers a more precise, efficient and scalable method for automated detection of diseases of cotton crops. We have implemented the proposed method on publicly available dataset with seven disease and one health classes and have achieved highest accuracy of 95% and F-1 score of 98%. The proposed method performs better than existing state of the art methods.

Introduction

Cotton is one of the most popular products in the textile market of the world and it has an important place in agriculture in the modern world (Ahmad & Hasanuzzaman, 2020). Cotton is widely known as ‘White Gold’ or the ‘King of Fibers’ and occupies the top most position among all the cash crops on which the textile industry completely relies as it is the major textile input. It sustains the livelihoods of around sixty million people and offers the economic profits to millions of farmers in both developed and the developing world (Prashar, Talwar & Kant, 2019). However, besides being a fiber crop, cottonseeds are also used as feed crop for animals, bio energy, staple and oil seed harvester and it contributes 35% of the global fiber and raw material that is used for textile (Zhang et al., 2013). Cotton is among the most preferred products in trade, yet its production is susceptible to invasion by various diseases pathogens (Kumar et al., 2021). Another challenge facing the growth of cotton is pests and diseases since they reduce the amount and quality of cotton produced which in turn affects the farmer’s revenue (Khan et al., 2020). Conventional pest and disease diagnosis relies on farmer perceptions and morphological assessment, which is cumbersome, labor-intensive, skill dependent, and not suitable for large scale commercial farming (Ul-Allah et al., 2021).

Advancements in deep learning and computer vision offer new opportunities for the automated detection of diseases and pests in agriculture (Rothe & Rothe, 2019; Stephen, Arumugam & Arumugam, 2024; Song et al., 2023). It helps in identification of different types of pests and diseases in the field images of the cotton plant and their location hence increasing the efficiency of pest and disease management (Caldeira, Santiago & Teruel, 2021). In the large scale farming this will minimize the labour requirements and increase the effectiveness and accuracy of pest and disease identification (Shah & Jain, 2019).

Lesions detection on cotton leave is very difficult in large areas of farmland and with lesions of different sizes due to the large amount of time and the low accuracy of manual detection. Automated detection methods have potential but also face challenges like dealing with dissimilar characteristics of data and minimizing false alarms. Our research addresses these challenges by effective preprocessing including data augmentation using GANs and an ensemble classifier after feature level fusion of features extracted from VGG16, Inception V3, and ResNet50. To improve the lesion detection performance, we perform the channel and spatial fusion of these models by means of image processing. In this study, a hybrid ensemble technique is employed to combine outputs of individual models leading to a reliable classification system. The proposed system will help farmers in enhancing the cotton yields and disease control and, at the same time, points to the importance of more studies in the context of cotton leaf lesion detection. Figure 1 shows RGB images of multiple diseases of cotton crop.

Figure 1 (A) Gossypium Hirsutism, (B) Leave Curl, (C) Root Rot, (D) Bollworm, (E) Foliage Leave, (F) Cotton Wilt, (G) Leaf Rol (Dhamodharan, 2023; Bhoi, 2020).

Dataset license type: ODbL V1.0.

Related work

A deep learning model in identifying plant diseases makes diseases more accurately identifiable. Identifying leaf diseases might pose challenges in the absence of prior knowledge. A deep residual neural network (DRNN) was offered as a solution by Vignesh, Askarunisa & Abirami (2023) and a plain convolutional neural network (CNN) to identify several cassava plant diseases. From Kaggle, they acquired a dataset. The accuracy of the author’s DRNN was 96.75%. The authors employed CNN to discern between healthy and sick leaf pictures on plant leaves using an online available dataset for diagnosing plant illnesses (Gavahi, Abbaszadeh & Moradkhani, 2021). Among other image processing methods, the findings included grab cuts, LBP, and image segmentation. In order to spot the pest, Talaviya et al. (2020) make use of a cutting-edge computational framework, including graphics processing unit (GPU) integrated processors and a deep residual network that utilizes a residual map. The authors employed deep learning techniques to facilitate the diagnosis of plant diseases. To accomplish this task, a deep convolutional neural network was utilized. The four most commonly utilized subclass algorithms in deep learning are CNN, long-short-term memory (LSTM), and visual geometry group (VGG). CNN, generative adversarial networks (GANs), and recurrent neural network (RNN) represent the three predominant Deep Learning methodologies in the present state of use. Nevavuori, Narra & Lipping (2019) describe utilized deep learning to examine vein patterns to find plant illnesses. The author hasn’t considered the color or form of the picture. Conventional CNN outperformed the anticipated machine vision pipeline. Supporting data is required in work to create a generic strategy for identifying all disorders, according to the author. The authors applied level one and level two segmentation methods to extract leaf-spot lesions (Cheung et al., 2023).

Many features were extracted, including border, color, and texture. AR-GAN was used for data augmentation to produce more images. The results gained 96.11% accuracy of the proposed strategy. Tran et al. (2019) presented a Deep CNN to identify various diseases in cucumbers. They used more than 1,000 images to identify the cotton crop disease in their suggested framework. The initial dataset included 14,208 tuples, and these were increased through data augmentation and achieved an accuracy of 93.4%. The less accuracy of the suggested model and the absence of training on additional benchmark datasets are the limitations. To identify several grape types, the author of the research (Fenu & Malloci, 2021) recommended utilizing Mask R-CNN. As a visualization technique, the researchers recommends utilizing salience maps to better comprehend and analyze CNN categorization. The authors make the case that the application of deep convolutional neural networks makes the diagnosis of cotton leaf disease easier. Li, Lecourt & Bishop (2018) offered a strategy for finding and classifying leaf lesions by genetic examination.

Transformer model is gaining popularity for handling sequential data (Vaswani et al., 2017). For cotton leaf lesion detection transformer model use attention mechanism which aids in assessing long term dependencies (Tay et al., 2021). Transformer model attention mechanism consider impact of each element in the input and generate reliable results. Mathematically it can be expressed as:

(1) Attention(Q,K,V)=softmax(QKTdk)V

where Q (Query), K (Key), and V (Value) represent the query, key, and value respectively, and dk denotes the dimension of the key. Knowledge Graph is another useful techniques for cotton lesion detection. Knowledge Graph is renowned technique for amalgamation and usage of specialized knowledge (Chen, Jia & Xiang, 2020). A knowledge graph G for cotton pest detection can be represented as:

(2) G=(V,E)

where V is the set of vertices (entities) and E is the set of edges (relationship between entities).

IoT is another evolving tool and technique for cotton pest detection. Modern devices and sensors intelligently capture and detect the cotton lesion. By using IoT platform farmer not only detect the cotton leaf lesion but also assess the soil condition and take further step accordingly (Patil & Patil, 2021).

Gülmez (2023) use Grey Wolf optimization algorithm for cotton lesion detection aids in dimensionality reduction and also enhance the accuracy of pest and disease detection.

(3) X→i(t+1)=X→i(t)−A→1⋅(C→1⋅X→α(t)−X→i(t))

where: X→i(t+1) is the updated position of the i-th wolf at time t+1,

X→i(t) is the current position of the i-th wolf at time t,

X→α(t) is the position of the alpha wolf at time t,

A→1 and C→1 are coefficient vectors,

A→1=2⋅a→⋅r→−a→ and C→1=2⋅r→,

a→ is a linearly decreasing vector from 2 to 0,

r→ is a random vector in the range [0, 1].

The author used several image processing algorithms on photographs of leaves taken specifically for the study. In order to reduce undesirable distortion, a smoothing filter is used after the masked cells have been removed and the threshold for green pixels has been applied. The best samples are chosen in each cycle after the samples are compared to the population. Table 1 presents the comparison of existing methods of automated cotton crop disease detection methods. The concept of clustering is adopted to extract different features of the image, including texture and hue. Leaf-spot lesions were extracted using two-stage segmentation (Hasanaliyeva et al., 2022). The researcher proposed employing another approach called deep convolutional network for the assessment of several distinct cucumber family spots. The study was conducted using an aggregate of 1,232 images. The presented dataset has 14,208 tuples and these were increased through data augmentation (Lee et al., 2019). According to the author, accuracy rates are 93.4%. The poor model accuracy and lack of training dataset laid a poor or mis-classification of healthy and unhealthy leaf disease detection. The approach recommends using salience maps to much more precisely analyze and apply convolution neural network classification. The study recommended using Mask R-CNN as a visualization technique to identify numerous grape varieties. According to the author, it makes deep learning more transparent. The author suggested using a genetic algorithm in his research to categorize and identify leaf disease (Punia, Singh & Madaan, 2020).

Table 1 Comparative analysis of the existing literature on automated detection of cotton crop diseases.

Publish date	Authors	Accuracy	Preprocessing	Feature extraction	Classifier	Classes	Total no. of images	Limitation	Future work	
2024	Li et al. (2024)	93.70%	–	CFNet-VoV-GCSP- LSKNet-YOLOv8	Customized model	6 classes: Army worm, Bacterial blight, Cotton boll rot, Diseased cotton leaf, Healthy, Target spot	4,703	Limited dataset size, Data imbalance, Evaluation metrics not given	Multi-channel scale attention mechanism, Enhancing detection efficacy, Model optimization	
2024	Nazeer et al. (2024)	99%	Resizing, Normalization, Agumentation	CNN	CNN	5 groups: Fully Resistant (FR), Partially Resistant (PR), Healthy (H), Partially Susceptible (PS), Fully Susceptible (FS)	1,349	Dataset size is limited, latest classifiers for better results.	Longitudinal data testing, increase the size of dataset, Comparative analysis of machine learning models.	
2024	Gao et al. (2024)	94%	Standardized images sizing, Normalization.	CNN, Transformer, Fusion model with knowledge graph	Transformer technology	10 classes: Foliar disease, Curl Leaves, Bacterial blight, Constrictor aphid, Red spot, Alternaria leaf spot, Herbicide, Spodoptera litura, Cotton bollworm, Cotton aphid	7,956	Model couldn’t handle larger dataset, Computationally complex model, Model couldn’t work under extreme environment condition.	Model optimization, Limited dataset size, Model adaptability.	
2024	Rasheed et al. (2024)	99.55%	Resizing, Rescaling, Normalization.	VGG16, Transfer learning.	Xception model	02 classes: Diseased and healthy class	946	Lightening issue while capturing images, Limited size dataset, Complex model for small dataset.	Optimization of model, increase the size of dataset, Field testing and validation.	
2024	Kukadiya et al. (2024)	98.13%	–	CNN, Inception V3	CNN, Inception V4	04 classes: bacterial blight, curl virus, fussarium wilt, healthy class.	1,786	Limited dataset size, Data imbalance, Evaluation metrics not given.	Expansion of dataset, testing of other classifiers results, generalizability of model.	
2024	Bhujade & Sambhe (2024)	96.07%	Gaussian filter, Segmentation.	Grey Level Co-Occurrence Matrix (GLCM), Local Directional Pattern (LDP), Global paatern	Hybrid model: CNN and Multi Resolution Feature Optimization (MRFO).	10 classes: Aphids, Armyworm, Bacterial blight, Healthy leaf, Powdery mildew, Target spot, Diseased cotton leaf, Diseased cotton plant, Fresh cotton leaf, Fresh cotton plant Herbicide, Spodoptera litura, Cotton bollworm, Cotton aphid	1,000	Pre-processing challenges, Limited feature extraction strategies, Time-consuming model.	Dataset Consumption, Advance pre-processing and feature extraction techniques.	
2024	Manurkar et al. (2024)	94.33%	Image resizing and standardization, Noise reduction filter, Contrast and brightness adjustments, Conversion to gray scale.	Color space analysis, Color histograms, Color filters, Edge detection using canny and sobel detector.	CNN	06 classes: Aphid, Army worm, Powdery mildew, Bacterial blight, Target spot, Healthy class.	3,600	Comparison of different classifiers were not given, Sensitivity and Specificity missing, Dataset variability issue.	Expansion of dataset, Test other machine learning models, Advance image processing techniques like morphological segmentation, pattern matching.	
2024	Lakkshmi Yogesh et al. (2024)	Orignal datset 98%, synthetic dataste 99%.	Normalization, resizing, augmentation, Noise reduction, Data balancing.	Transfer learning, CNN, Feature maps.	Mobile Net V2	04 Classes: Healthy, Bacterial blight, Curl virus and fusarium wilt.	1,711	Limited diversity in diseased leaves, Evaluation metrics missing, Only CNN models were assessed.	Dataset Expansion, Explore novel architecture, Ensemble learning, Transfer Learning strategies, longitudinal studies.	
2024	Thivya Lakshmi, Katiravan & Visu (2024)	–	–	High Resolution Technology, CNNs, Dimensionality Reduction, Point Cloud Processing, Nearest Neighbour Method.	96%	–	30,000	Limited dataset size, The accuracy, sensitivity, specificity and no of classes were not mentioned.	Multi-class classification, Validation and comparitive study, Limited size of datasets, Assess variability.	
2024	Abdalla et al. (2024)	89.70%	Image segmentation, Handling class imbalance,	CNN, Hand-crafted Features (vegetation fraction, color and texture)	Resnet101- BiLSTM Model	02 classes:fusarium oxysporum disease class and healthy class	3,684	Model optimization required, Evaluation metrics missing, Comparative analysis of multiple classifiers were not given.	Address class imbalance issue, Consider environmental factors, Temporal data analysis, Modal optimization.	
2024	Yang et al. (2024)	90.62%	Continuous wavelet transform (CWT), Removed outliers.	Continuous wavelet transform (CWT), LASSO-VIF (Variance inflation factor), Wavelet features.	Logistic Regression Model	02 classes: Verticillium wilt and healthy class.	–	Costly technique, Complex Methodology, Laborious task, Time consuming task, Methodology not suitable for large fields.	Increase the size of dataset, Less time-consuming strategy required, Comparitive analysis of other machine learning models.	
2024	Hyder & Talpur (2024)	98.50%	Cleaning using a mean filter, Image segmentation.	CNN	Proposed Model	04 classes: Bacterial blight, Curly virus, Fussarium wilt and healthy leaves.	1,710	Only three classes are assessed, Dataset size is limited, Data Variability was not considered.	Enhance model performance, Improve dataset quality, Expand dataset size, Considered other classes for cotton disease detection.	
2023	Hyder & Talpur (2024)	98.45%,	Image filtering, Segmentation, Laplacian noise filter,	ResNet50 and customized model named as Advance Learning Model (ALM)	CNN	07 classes: Healthy, Leafspot, Nutrient deficiency, Powdery mildew, Target spot, Verticillium Wilt, and Leaf curl).	2,384	Sensitivity, Specificity were not presented, Class imbalance issue exist, Comparison with other models were not presented	Expansion of dataset, Integration of transfer learning, Developed new algorithms.	
2023	Yadav et al. (2023)	80%	Image resizing, Data annotation, Data agumentation.	YOLOv3	YOLOv3	02 classes: Diseased and healthy class	2,250	Only one disease was analyzed, Limited dataset, Evaluation metrices were not discussed.	Increase dataset size, Increase number of diseased classes Comparative analysis of latest classifiers, Consider Temporal data.	
2023	Gülmez (2023)	93%	Image agumentation and fine tuning.	CNN	Proposed Model optimized by Grey Wolf Algorithm	04 classes: Cotton leaf, Diseased cotton plant, Fresh cotton leaf, and Fresh cotton plant.	2,057	Small dataset size, Generalizability of model was not considered, Evaluation metrics were not mentioned.	Comparative analysis of novel algorithms with proposed model, increase dataset size, consider data variability, Study Temporal data.	

Proposed methodology

Proposed method consists of three steps including preprocessing, feature extraction and classification of RGB images for multiclass disease and healthy leaves of cotton crop. The framework is applied on customized publicly available dataset for cotton disease detection. This is followed by thorough preprocessing, feature extraction and feature fusion. The application of a CNN within the domain of disease classification presents innovative progressions in the realm of transfer learning. Figure 2 presents CNN architecture for automated lesion detection from cotton images. Pre-trained architectures are used in the ensemble method like the model VGG16, Inception V3 model, and ResNet50 algorithm to achieve consensus via majority voting, thereby amalgamating the individual strengths of each model. These three models were chosen for the ensemble because they are well-balanced in image properties handled according to their complementary strengths. We also perform additional experiments over various subsets of these architectures to validate this decision. As expected, the ensemble of all three was always superior to any two confirming a diverse ensemble is beneficial. Every model was assessed based on its accuracy, sensitivity, specificity, precision, and F1 score. Table 2 shows how various combinations of models performed in terms of accuracy and F1 score. Among various methods of aggregating outputs of multiple models, the simple majority voting was selected as a result of ensemble method since deep learning architecture is complicated, In addition, it is easier implementing and more reliable.

Figure 2 CNN architecture for automated lesion detection in cotton leaf visuals.

Table 2 Performance of different model combinations.

Model Combination	Accuracy (%)	F1 score	
VGG16 + Inception V3	95.2	0.94	
VGG16 + ResNet50	96.0	0.95	
Inception V3 + ResNet50	95.8	0.94	
VGG16 + Inception V3 + ResNet50	98.5	0.97	

The complexity advantage that majority voting holds for the ensemble eliminates the need for bringing additional measures. That is, this approach is especially useful when there is a high level of model differentiation because it allows the averaging of highly divergent predictions to give an increase in the general level of accuracy and model stability (Pan et al., 2020). The framework consolidates data to help farmers and experts quickly identify diseased cotton leaves. This novel method connects technology and agriculture, improving cotton crop disease detection. Figure 3 presents the proposed framework for cotton crop pest identification.

Figure 3 Proposed framework for cotton crop pest identification.

Preprocessing

During the preliminary stage of the proposed framework, collection of an extensive dataset comprising images of cotton crops are performed. Images are obtained from Kaggle datasets. Prioritizing the inclusion of a wide range of healthy and unhealthy cotton leaves plays significant role. For the data collected, 80% of it constituted training set while the other 20% was used for validation. This ensures a good amount of data for training the model as well as validation of its performance. In addition to the competitor (Rastogi et al., 2024), another thing that also has a big effect on how well disease detection models develop is dataset size and diversity. In this stage, several steps are executed. A Gaussian filter is widely used for image smoothness by applying the average on pixels when contrasting with its neighbour, and then the Gaussian function is used for weight calculation (Kaur et al., 2024; Paul Joshua et al., 2024). The Gaussian filter can be expressed by following equation:

(4) G(x,y)=12πσ2exp⁡(−x2+y22σ2)

where: G(x,y) is the Gaussian function.

x and y are the distances from the origin in the horizontal and vertical axes, respectively.

σ is the standard deviation of the Gaussian distribution.

Gaussian filter helps in making boundaries more visible also enhance the lesion detection on cotton leaf by improving the variability among the images and smooth the images by removing the high-frequency noise. For detection of edges in image Laplacian filter is applied, when rapid intensity change is observed in specific region Laplacian filter highlights that region (Ramu et al., 2024). The Laplacian filter when applied for cotton disease detection it filters out the uninterested region and make the interested region of image more prominent that create ease for deep learning model for lesion detection in cotton crop images (Lv et al., 2024; Yamada, 2024). The Laplacian filter can be expressed by following equation:

(5) Δf(x,y)=∂2f∂x2+∂2f∂y2

An example of a popular kernel for a 3 × 3 neighborhood Laplacian filter is:

(6) [0101−41010]

A pre-trained generative adversarial network (GAN) was used to produce synthetic images, generated from the dataset for minority classes. The GANs was trained specifically for the underrepresented classes to fix this issue of class imbalance in our real dataset. Figure 4 presents GANs-generated images. In GANs, two networks compete with each other and tries to discriminate between real and synthetic image. GANs play a pivotal role in synthesizing images that not only resemble real images but also increase the dataset size, robust model for lesion detection and improve the training process (Gangadhar et al., 2024; Zekrifa et al., 2024). GANs function can be expressed as following:

(7) minGmaxDV(D,G)=Ex∼pdata(x)[log⁡D(x)]+Ez∼pz(z)[log⁡(1−D(G(z)))]

where: D(x) represents the probability that the discriminator correctly identifies a real image, rather than the absolute probability that x is real.

z is a random noise vector.

G(z) is the synthetic image generated from z.

Figure 4 Images generated through GANs (A) Target spot, (B) Mildew, (C) Aphid, (D) Bacterial blight (Dhamodharan, 2023; Bhoi, 2020).

Dataset license type: ODbL V1.0.

Feature extraction

The process of feature extraction is one of the main elements, which “translates” raw data to more structured and interpretable form of the feature representation which can be comprehended by deep neural network models. This phase aid in identifying relevant features and pattern for the identification of cotton leaf lesion detection. Three different architectures: VGG16 is used for relevant pattern extraction. VGG16 plays significant role for the extraction of features due their simplicity, depth, hierarchical feature extraction, spatial dimensionality reduction and pre-trained weight enables transfer learning (Mohmmad et al., 2024). The convolution functioning for VGG16 can be expressed using following equation:

(8) Fl+1(x,y)=σ(∑i=−kk∑j=−kjwijFl(x+i,y+j)+b)

where: Fl is the feature map at layer l,

wij are the convolutional weights,

b is the bias,

σ is the ReLU activation function,

k is the kernel size.

The Inception V3 module used parallel convolutions for the extraction of features from cotton leaf in order to identify lesion. The Inception V3 module increase computational efficiency, and ensure the reduction of over-fitting due it hierarchical nature of feature extraction and transfer learning capabilities (Meena et al., 2023). The feature extraction process in the Inception V3 module can be represented by following equation:

(9) O=[f1×1(I),f3×3_reduce(I),f3×3(I),f5×5_reduce(I),f5×5(I),ppool(I)]

where: I is the input feature map,

f1×1, f3×3_reduce, f3×3, f5×5_reduce, and f5×5 are convolution operations with respective kernel sizes,

ppool is a pooling operation.

ResNet50 freeze top layers and popular for feature extraction due its residual connections, transfer learning capabilities and generalization ability. When compared with traditional feature extraction techniques, ResNet50 can capture more abstract and complex patterns. The following equation expressed the extraction of patterns using ResNet50.

(10) y=F(x,{Wi})+x

where: x is the input feature map,

y is the output feature map,

F is the residual mapping to be learned,

{Wi} are the weights of the layers in the residual block.

Feature fusion

In this phase features that are extracted using VGG16, Inception V3 and ResNet50 fused together using channel and spatial attention. For adaptability of context, reduction of model complexity and accurate lesions variations detection channel and spatial attention feature fusion technique is used. In channel attention mechanism weight for each feature is computed. These computed weights are then applied to each channel in order to eliminate irrelevant channel. The mathematical representation of channel attention weight calculation is as follows:

(11) Mc(F)=σ(MLP(AvgPool(F))+MLP(MaxPool(F)))

where: Mc(F) is the channel attention map,

F is the input feature map,

σ is the sigmoid activation function,

MLP is a multi-layer perceptron,

AvgPool and MaxPool are average and max pooling operations, respectively.

Spatial attention mechanism is used for understanding of relationship among spatial space and ignore the irrelevant regions. Weight is computed for each spatial region for the selection of informative regions by using following mathematical expression.

(12) Ms(F)=σ(Conv1×1([AvgPool(F);MaxPool(F)]))

where: Ms(F) is the spatial attention map,

F is the input feature map,

σ is the sigmoid activation function,

Conv1×1 is a 1×1 convolution operation,

AvgPool and MaxPool are average and max pooling operations, respectively.

When features are extracted using spatial and channel attention mechanism then these features are fused together by multiplying each element of input feature map. Thus ensure the better performance and reliable results for combining the relevant features from cotton leaves. Following mathematical equation express the feature fusion mechanism.

(13) Ffused=F⊙Mc(F)⊙Ms(F)

where: Ffused is the fused feature map,

F is the input feature map,

Mc(F) is the channel attention map,

Ms(F) is the spatial attention map,

⊙ denotes element-wise multiplication.

Classification

During the post-processing step, the conclusions made by the multitude of ensembles are aggregated in order to determine the health status of the cotton leaf. Ensemble learning involved the use of three correct models of VGG16, Inception V3 and ResNet50 which ensured accurate classification of cotton lesion. This disease categorization is determined by attracting the highest average number of votes among all the corresponding models. The application of this self-organised decision making method helps to reduce prejudices and improve the overall robustness of the model even despite possible inaccuracy of the individual forecasts. The increase in accuracy of the decision is due to the fact that, various models can be trained either on different sub-routines of training data or different hyper- parameters. Thus, using multiple models promises to achieve a higher level of accuracy due to the Law of Large Numbers and the Central Limit Theorem, although, at the same time, one must admit that the assumption of independent and identically distributed (i.d.d.). Which most likely does not apply to deep learning’s outputs. Due to the variety of the architectures of the models and training strategies the further forming of their outputs is not independent identically distributed (Zhou, 2012). However, more variations in this case help to decline the phenomena of over-fitting and cover more features. Other reasons for environmental potency of the ensemble are the established stability arising from the different angle visions and the option for avoiding biases of individual models. From a theoretical perspective, the convergence of forecasted probability distributions toward the consensus prediction can be described mathematically using the principles of the Law of Large Numbers and the Central Limit Theorem (Lipnik, Madritsch & Tichy, 2024). Let X1,X2,…,XN denote the outputs of these models, which are treated as random variables, and SN represent the collective response provided by the models. As the number of models increases, the relative sample mean SN converges to the true sample mean in almost all samples according to the Law of Large Numbers. This convergence is expressed mathematically as:

(14) limn→∞P(|1n∑i=1nXi−μ|<ε)=1

where: μ represents the true underlying disease class,

ε is a small positive number representing the desired level of precision.

When the number of models increased this lead towards infinity so the variable Sn merge to a normal distribution. Class with highest probability represents the level of dispersion of the decision made by the ensemble. Due to the central limit theorem which claims distribution of the sample mean converges to a normal distribution as the sample size gets bigger (del Barrio, González Sanz & Loubes, 2024).

(15) n(1n∑i=1nXi−μ)→dN(0,σ2)

where: σ2 is the variance of the individual model predictions, and

→d denotes convergence in distribution.

The code of the methodology is available at https://doi.org/10.5281/zenodo.13324708

Results and discussion

Dataset

We developed a custom dataset by combining images from different publicly available datasets on Kaggle. The dataset comprises 1,300 images. Eight classes are identified: Aphid, Army Worm, Blight, Healthy, Leaf Curl, Mildew, Leaf Spot, and Wilt. We applied augmentation to balance the dataset and increase the number of images. After augmentation the number of images is 4,793. Table 3 presents the number of images in each class, both before and after augmentation. Table 4 presents the total parameters, Flops (floating-point operations), and training time for VGG16, Inception V3, ResNet50, and our proposed model.

Table 3 Description of dataset composition.

Class	Number of images	
Without augmentation	
Aphid	100	
Army worm	150	
Blight	200	
Healthy	250	
Leaf Curl	120	
Mildew	130	
Leaf spot	180	
Wilt	170	
Total	1,300	
With augmentation	
Aphid	580	
Army worm	550	
Blight	600	
Healthy	750	
Leaf Curl	620	
Mildew	590	
Leaf spot	580	
Wilt	523	
Total	4,793	

Table 4 List of parameters of different deep learning models.

Model	VGG16	Inception V3	ResNet50	Proposed model	
Total parameters	918,566	1,487,591	25,638,875	2,629,639	
FLOAPs	5.1G	30.9G	0.00421G	38.5G	
Training time	1 h 3 m 56 s	54 m 3 s	51 m 5 s	1 h 17 m 2 s	

Performance measure

Each model was evaluated using accuracy, sensitivity, specificity, precision, and F1 score. The model’s accuracy is established by mathematical formulae.

(16) Accuracy=TP+TN(TP+FN)+(FP+TN)

(17) Sensitivity=TPTP+FN

(18) Specificity=TNTN+FP

(19) Precision=TPTP+FP

(20) F1Score=2×TP2×(TP+FN+FP)

Experimental results

Deep convolution networks (DCNs) come across as autonomous learners that are able to derive features from pre-learned images that have passed through discernment of the images via a reliable protocol. This step enhances the model’s capacity to identify distinct patterns and features present alongside healthy and diseased cotton leaves. DCNs present themselves as unsupervised learners which have the ability to extract features from pre-learned images that went through a process of discerning them based on some trusted method. This step helps the model differentiate between different patterns and properties that are observed in healthy vs. diseased cotton leaves.

Many deep neural network methods have been proposed to identify several leaf diseases. These methods include forming custom deep neural networks (DNNs), but also, reusing existing pre-trained nets. In our study, we utilized the Kaggle dataset to conduct an analysis that involved the application of various deep learning models, namely Inception V3, ResNet50, VGG16, and a Transfer Learning approach. This led to the development of a comprehensive ensemble of pre-trained models through training procedures. Datasets used in this study are publicly available at: https://www.kaggle.com/datasets/dhamur/cotton-plant-disease/data and https://www.kaggle.com/datasets/janmejaybhoi/cotton-disease-dataset/data.

By incorporating a diverse range of models, there is a potential to encompass a broader array of leaf attributes compared to relying solely on a singular paradigm. Inception V3, VGG 16 and ResNet 50 results are combined on the bases of voting in order to extract a wide range of leaf features from a comprehensive integrated model. Table 5 summarizes the comparison FLOPs and training time of each model and proposed ensemble based learning model. ResNet50 has the least FlOPs and training, yet successfully achieved highest accuracy when compared with VGG16 and Inception V3. ResNet 50 is beneficial for real time applications due to its residual blocks where computing power is the constrained (Falaschetti et al., 2021; Anand et al., 2024). Table 6 presents the Precision, Recall, and F-score results of the proposed model. Adding more models to the ensemble improves its performance, as seen by higher accuracy and F1 scores. This strategy exhibits a remarkable capacity for making sweeping generalizations.

Table 5 Comparison of FLOPs and training time.

Model	Total parameters	FLOPs	Training time	
VGG16	918,566	5.1	1 h 3 m 56 s	
Inception V3	1,487,591	30.9	54 m 3 s	
ResNet50	25,638,875	0.00421	51 m 5 s	
Proposed model	2,629,639	38.5	1 h 17 m 2 s	

Table 6 Precision, recall and F1 score achieved using proposed method.

Parameters	Precision	Recall	F-Score	
Aphid	0.92	0.84	0.91	
Army worm	0.92	0.92	0.97	
Blight	0.90	0.94	0.94	
Healthy	0.93	0.93	0.93	
Leaf curl	0.90	0.90	0.89	
Mildew	0.94	0.84	0.94	
Leaf spot	0.93	0.98	0.97	
Wilt	0.95	0.93	0.98	

The model was trained and tested on deep neural network and afterwards fine-tuning was performed by applying hyper-parameters. The optimizer used for ResNet50 was SGD and the rest of models were trained using Adam. We made this decision based on the specific properties and training dynamics of ResNet50, for which SGD with momentum has been proved to be beneficial in terms model convergence and generalization. Different optimizers were employed, so that in each case the strength of models architecture was utilized for a training process.

Table 7 presents hyper parameters for Inception V3, VGG16, ResNet50 and the proposed model. The model was trained in 1 h, 17 min, and 2 s. Five stages of convolution, a dropout layer, and a max-pooling layer are incorporated into the model of deep neural networks to boost performance. The softmax layer has eight classes for diagnosing leaf disease. Proposed model employed a total of 2,629,639 parameters, resulting in an accuracy rating of 98.5% as given in Table 4. Transfer learning is applied on VGG16, Inception V3, and ResNet50 models. It was the best response to an ILSVRC challenge. VGG16 contains a total of 16 layers. We conducted our experiment by freezing the VGG16 pre-trained model’s outermost layers. The model was executed in 54 min and 3 s on a cotton dataset with 18,678,382 parameters, and it had a 91.33% accuracy. In comparison to a Inception V3, the model had an accuracy that was significantly higher while training on fewer parameters. An already trained machine learning model is called ResNet50. On the Image-net dataset, the model was initially trained. To put it into practice, ResNet50’s top layers were frozen. A total of 95.12% of the model’s predictions were accurate. Whereas Inception V3 model got 89.61% accuracy. In ensemble technique, voting was performed and on the basis of majority voting disease was labeled with a bounding box. Figure 5 represents results achieved from three different architectures for lesion detection.

Table 7 Hyperparameters of models.

Model	Name	Parameters	
VGG-16	Optimizer	Adam	
	Learning rate	0.001	
	No. of epochs	100	
	Dropout	0.5	
	Batch-size	32	
	Input shape	224 × 224 × 3	
Inception V3	Optimizer	Adam	
	Learning rate	0.001	
	No. of epochs	100	
	Dropout	0.5	
	Batch-size	32	
	Input shape	299 × 299 × 3	
‘ResNet50	Optimizer	SGD	
	Learning rate	0.001	
	Weight decay	0.001	
	Momentum	0.9	
	No. of epochs	100	
	Dropout	0.4	
	Batch size	32	
	Input shape	224 × 224 × 3	
Proposed model	Optimizer	Adam	
	Learning rate	0.001	
	No. of epochs	100	
	Dropout	0.25	
	Batch-size	32	
	Input shape	224 × 224 × 3	

Figure 5 Comparison of results achieved using different experimental approaches.

The ratios of true positive (TP) and true negative (TN) effectively distinguished an equivalent proportion of imagery depicting healthy and aberrant leaves, respectively. To describe instances of healthy leaves that are incorrectly labeled as sick, and vice versa, the terms “false positive” (FP) and “false negative” (FN) are used. To determine the confidence range for the accuracy of classification, it is important to calculate the ratio of correct predictions to the total number of predictions. The model’s efficacy is assessed by considering the F1 score, precision, and recall, along with the traditional metrics for assessment. Table 6 presents the Precision, Recall, and F-score results of the proposed model. Based on the data depicted in Fig. 5, When compared to other models on our cotton dataset, our suggested method achieves a remarkable 98.50% accuracy. Figure 6 depicts a heatmap representing each class.

Figure 6 Confusion matrix.

The above-mentioned result demonstrates the model’s capability to successfully navigate the intricate domain of cotton disease detection, indicating encouraging implications for the field of precision agriculture.

Conclusion

The growing number of plant diseases in the tropical agriculture has made it necessary to use the latest technology to shield the crops from the recurrent outbreaks. DNNs are the best solution that are more efficient than the conventional methods. Our method is based on the combination of various ensemble techniques in a single DNN system, thus, providing a holistic and cost-efficient way of dealing with crop problems. VGG16 is the improved deep neural network combined with Inception V3, ResNet50 with image pre-processing, as used for channel and spatial integration. This lead to the formulation of a firm ground classification system that will in a way benefit the farmers in enhancing yield as well as blood diseases patent. The strength of large dataset which is combination of both healthy images and images with lesions is very helpful to address challenge created by variability of training context to the actual practice environment. Our method gave very high and accurate percentage of 98% demonstrating our method efficient used in fighting plant diseases and provides the opportunity to build a new ‘Green Digital Agronomy’ for the cotton’s sector.

For detection of cotton leaf disease an ensemble method gives promising results. This is where technology comes into play, to help farmers in a practical manner so they can track and manage crop health more effectively. With this system in place for regular activities, farmer can detect disease outbreaks rapidly and will be able to identify the issue beforehand resulting increased crop yield rate thereby minimizing damages crops due to diseases. Future work will include model improvements and generalisation to other crops. By deploying a deep learning model on mobile applications or farm management software real time disease detection and diagnostics can be achieved. The use of such tools results in farmers getting valuable insights into the exact status on field which helps them plan timely interventions and manage crops better. The quest for innovation in crop production is the key to the solution of future problems and the world’s health. Ongoing research and innovation will make our model become more improved and evolved to provide practical solutions to the agricultural problems and at the same time to be a step for modern farming.

Additional Information and Declarations

Competing Interests

Author Contributions

Data Availability

The authors declare that they have no competing interests.

Frnaz Akbar conceived and designed the experiments, performed the experiments, analyzed the data, performed the computation work, prepared figures and/or tables, and approved the final draft.

Yassine Aribi conceived and designed the experiments, analyzed the data, performed the computation work, prepared figures and/or tables, and approved the final draft.

Syed Muhammad Usman conceived and designed the experiments, performed the experiments, analyzed the data, performed the computation work, prepared figures and/or tables, authored or reviewed drafts of the article, and approved the final draft.

Hamzah Faraj performed the experiments, performed the computation work, prepared figures and/or tables, authored or reviewed drafts of the article, and approved the final draft.

Ahmed Murayr performed the experiments, analyzed the data, performed the computation work, authored or reviewed drafts of the article, and approved the final draft.

Fawaz Alasmari analyzed the data, performed the computation work, authored or reviewed drafts of the article, and approved the final draft.

Shehzad Khalid conceived and designed the experiments, performed the experiments, analyzed the data, prepared figures and/or tables, authored or reviewed drafts of the article, and approved the final draft.

The following information was supplied regarding data availability:

The cotton plant disease data is available at Kaggle:

https://www.kaggle.com/datasets/dhamur/cotton-plant-disease/data, DOI: 10.34740/kaggle/dsv/5127834.

The Cotton Disease Dataset is available at Kaggle: https://www.kaggle.com/datasets/janmejaybhoi/cotton-disease-dataset/data.

The code is available at GitHub and Zenodo:

- https://github.com/FrnazAkbar/Cotton-Lesion-Detection/tree/991640ddd25ad2fee85ee41f1bc92d1ea406a55b

- FrnazAkbar. (2024). FrnazAkbar/Cotton-Lesion-Detection: Automated Lesion Detection in Cotton Leaf Visuals using Deep Learning: Code Release (v1.0). Zenodo. https://doi.org/10.5281/zenodo.13324708.

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
