# Peer review of "Automated lesion detection in cotton leaf visuals using deep learning"

_PeerJ Computer Science, doi:10.7717/peerj-cs.2369_

## Round 0.1 · original submission · Major Revisions

· Academic Editor

Major Revisions

The paper must be improved as per the comments of the reviewers.

·

Basic reporting

The manuscript is well-written with clear and professional English. The literature review is comprehensive, and the paper structure with figures is professional and easy to follow.

Experimental design

The research question is well-defined and relevant, focusing on cotton disease detection with an ensemble-based deep learning model. The methodology is described in detail, allowing for replication.

The paper would benefit from a discussion on why a traditional ensemble method (majority voting) was chosen over a deep learning architecture for combining model outputs.

Validity of the findings

The experiments are well-designed and conducted, and the conclusions are sound based on the presented results.

The authors utilize the Law of Large Numbers and the Central Limit Theorem to explain the ensemble's potential benefits (lines 165-175). However, this explanation needs revision. While the concept of leveraging multiple models for improved accuracy aligns with these theorems, the i.i.d. (independent and identically distributed) assumption likely doesn't hold for the outputs of different deep learning models. The authors should clarify this limitation and potentially discuss alternative justifications for the ensemble's effectiveness.

A minor confusion exists in the explanation of GAN function in Equation 7 (lines 125-129). The text suggests D(x) should always be 1 if x is a real image. Explain that D(x) should represent the probability of the discriminator correctly identifying the real image, rather than the absolute probability that x is real.

Additional comments

The paper presents a well-motivated and executed approach to cotton disease detection using an ensemble-based deep learning model. The authors should address the comments regarding ensemble method choice and the limitations of the statistical explanation.

Overall, this is a well-written paper with a strong foundation. Addressing the minor points mentioned above will further strengthen the manuscript.

·

Basic reporting

no comment

Experimental design

The authors have taken on a very interesting ensemble approach combining different DL architectures to do cotton diseases multi-classification. However, some more clarity is needed to show the robustness or reproducibility of this study:

1. in figure 3, it shows the dataset contains 8 classes, but in table 2 only lists the number of healthy vs unhealthy leaves. It'd be much appreciated to show the composition on each of those 8 classes which gives an idea of data imbalance.

2. For the GAN based argumentation, I'm not too sure how this was done. Did the authors take a pre-trained GAN and directly apply it or train one from the scratch? Since it is used to upsample minority classes, what classes, and how many of it did the GAN generate? This is not shown in the result. It'd be also nice to show a couple of GAN-generated images just to see the quality and consistency. In order to generate images for specific classes, I believe we needed to use a conditional GAN; labels for the generated images wouldn't be known otherwise.

3. The author did a train validation split for the training, which is nice. What is the ratio of the split it is done on? You have to report this for reproducibility purpose.

Validity of the findings

It is not clear to me why these specific 3 architectures (VGG, Inception V3 and Resnet 50) were chosen as an ensemble. In order to show this, I'd suggest the authors to do a few more subsetting and combinations. Take 2 out of those 3 models to show if really is the more the merrier. In addition, in spite of the good performance, it is not convincing that this ensemble method is superior to the existing ones from the literature as the underlying dataset is different. Some benchmark experiments on the same dataset should be conducted.

Reviewer 3 ·

Basic reporting

The paper is written in clear, unambiguous, professional English, effectively communicating the complex methodologies and results. However, there are minor grammatical errors and awkward phrasings that could benefit from thorough proofreading.
1. In Table 3, the ResNet 50 has the lowest FLOAPs compared to other models, and it used the least amount of training time with the second-highest accuracy (Figure 4). The authors should discuss the differences in the manuscript.
2. I noticed that the authors used SGD as the optimizer for ResNet 50, but Adam for the rest. Is there a particular reason?
3. The authors could expand the conclusion more to how the farmers could apply this technology to daily farming.

Experimental design

The study fits well within the aims and scope of the journal, focusing on the application of AI in agriculture. The objectives are clearly stated: to develop a precise, efficient, and scalable method for cotton leaf lesion detection using deep learning. The proposed method involves an ensemble of three deep neural networks: VGG16, Inception V3, and ResNet50. This approach is well-justified given the strengths of each model in handling different aspects of the image data. The use of image processing techniques for enhancing picture quality and lesion details is a notable strength. The methodology is described with sufficient detail to allow replication, including equations and explanations of key processes like feature extraction and fusion. Data preprocessing steps, including Gaussian and Laplacian filters and data augmentation using GANs, are well-detailed. These steps are crucial for improving model performance and generalizability. The evaluation metrics used (accuracy, sensitivity, specificity, precision, F1 score) are appropriate for assessing the performance of classification models. The inclusion of a confusion matrix and various performance graphs adds to the robustness of the evaluation.

Validity of the findings

The paper demonstrates a significant improvement over existing methods, particularly in handling varied lesion sizes and different data attributes. The hybrid ensemble approach is innovative and provides a more robust solution compared to single-model approaches. The study encourages replication by providing detailed descriptions of the dataset, preprocessing steps, and model architectures. However, it would be beneficial to include a link to the dataset and code repository to facilitate reproducibility. The conclusions are well-supported by the results, emphasizing the effectiveness and scalability of the proposed method. The paper identifies future directions, such as further improving model performance and exploring additional datasets.

Additional comments

1. Line 39: I don’t think any animal would be fed by cotton, but cottonseeds. Please be specific.
2. The authors could consider combining figures 5-8.
3. Table 5, the input shape for the proposed model is missing.

Cite this review as

---

## Round 0.2 · accepted · Accept

· Academic Editor

Accept

The paper can be accepted. It was well improved.

·

Basic reporting

The author has maintained clear and unambiguous, professional English throughout. Literature references, field background, and context are sufficient. Article structure, figures, tables, and raw data sharing are professional.

Experimental design

The research question is well-defined and relevant. Methodology is described in sufficient detail to replicate. The author has addressed the concern about the ensemble method choice.

Validity of the findings

Experiments are well-designed and conducted. Conclusions are sound based on presented results. The author has revised the explanation of the ensemble's benefits and clarified the GAN function.

Additional comments

The author has addressed all my concerns. I have no more comments.

Reviewer 3 ·

Basic reporting

The authors addressed my comments carefully and the manuscript is in good shape for publication. I recommend to accept and publish the paper.

Experimental design

The authors addressed my comments carefully and the manuscript is in good shape for publication. I recommend to accept and publish the paper.

Validity of the findings

The authors addressed my comments carefully and the manuscript is in good shape for publication. I recommend to accept and publish the paper.

Cite this review as